# BFCL Audio: An Audio Function Calling Evaluation for Large Language Models

**Huanzhi Mao** [1] **Aditya Ghai** [1] **Imra Dawoodani** [1] **Tony A Ginart** [2] **Shishir G Patil** [1] **John Emmons** [2]
**Joseph E. Gonzalez** [1]

## Abstract

Audio agents are increasingly deployed to execute tools from spoken requests, yet audio tool use poses challenges beyond text-only function calling: perception errors (e.g., homophones, noise, disfluencies) can corrupt entities and arguments, and natural interactions often require clarification that changes the tool-calling protocol. We introduce BFCL Audio, a large-scale benchmark for *audio function calling* with **6.2K** expert-verified tasks across two suites that mirror common deployments: BFCL Text Audio (pipelined ASR → LLM → tools via transcripts) and BFCL True Audio (end-to-end audio-in → tool calls). BFCL Audio includes controlled speech and acoustic perturbations (accent and speaking-rate variation, content disfluencies, and background noise) generated through a controllable audio synthesis/augmentation pipeline. We provide automatic grading for both function names and argument values using AST-based matching for single-turn calls and response/state-based metrics for multi-turn interactions, enabling scalable evaluation without LLM judges. Across a broad set of models, we propose a failure-mode taxonomy and analyze which speech and noise factors most strongly impact tool-calling accuracy. We release the benchmark, evaluation harness, and audio pipeline to support research on reliable speech-based agents.

## 1. Introduction

Large Language Models (LLMs) have rapidly transitioned from pure text interfaces to *tool-augmented* agents capable of calling external functions such as database look–ups, API endpoints, or robotic controllers. Recent releases from leading labs and open-source communities such as GPT 5,

Gemini 3.0 Pro, and Llama 4 have extended this capability beyond text: a single model can now listen and speak, invoking the same JSON-style function calls from raw audio. Despite the commercial excitement, we lack a systematic evaluation of how well existing audio-capable models actually perform *end-to-end function calling*.

Current benchmarks focus on either (i) text-only tool use, for example, BFCL(Patil et al., 2025), ToolBench(Qin et al., 2023), and $\tau$-BENCH(Yao et al., 2024), or (ii) general multimodal understanding such as MMMU(Yue et al., 2024). However, *audio* function calling is not a simple extension of text function calling: speech introduces new error channels (e.g., ASR mistakes under background noise, accent and speaking-rate variation, disfluencies, and homophones) that can silently alter entities and slot values. Moreover, spoken requests are often genuinely underspecified, so robust agents must be evaluated on *when* and *how* they ask for clarification—changing the interaction pattern from single-shot parsing to multi-turn repair. As a result, an audio benchmark cannot be "text benchmarks + TTS": it must explicitly model realistic acoustic perturbations and support clarification behavior in the evaluation protocol. Without such diagnostics, it is impossible to decide whether an error stems from perception, reasoning, or formatting and therefore impossible to improve the system in a targeted manner.

We introduce **BFCL Audio**, the first benchmark to fill this gap and evaluate end-to-end *audio* function calling (tool use). BFCL Audio contains *6.2K* diverse tasks across two evaluation suites: BFCL Text Audio and BFCL True Audio. These suites reflect two common production deployments: (i) a *pipelined* approach (ASR → LLM → tool calls), which BFCL Text Audio targets by providing transcripts, and (ii) an *end-to-end* approach (audio-in → tool calls), which BFCL True Audio targets by requiring direct inference from speech. Each task specifies (1) a user request (transcribed text and/or speech), (2) a sequence of ground-truth function calls, and (3) controlled distractors that stress different parts of the tool-calling pipeline, including background noise, accent variety, and content disfluencies. Moreover, BFCL Audio is built with a controllable audio generation and augmentation pipeline that can independently manipulate speech attributes and acoustic

[1]University of California, Berkeley [2]Salesforce AI Research. Correspondence to: Huanzhi Mao <huanzhimao@berkeley.edu>.

*Proceedings of the 43rd International Conference on Machine Learning*, Seoul, South Korea. PMLR 306, 2026. Copyright 2026 by the author(s).

conditions (and their strength), enabling realistic synthetic audio rather than naive TTS. We deploy an AST substring-matching metric for single-turn function calls and response-based and state-based metrics for multi-turn function calls, enabling automatic grading without relying on fragile LLM judges. "AST" refers to the Abstract Syntax Tree matching procedure used for single-turn evaluation. In the paper, single-turn scoring parses the predicted tool call into an AST and checks function-name and argument-level correctness without executing the function.

To better diagnose failures, we propose a failure-mode taxonomy tailored to audio tool use to pinpoint where tool use breaks. Our analysis uncovers two recurring failure modes: (i) **Incorrect Named-Entity Recognition**, where mis-hearing a single proper noun invalidates the call by generating an incorrect argument or function name; and (ii) **Clarification Failures**, where models ask unnecessary or irrelevant follow-up questions. We further quantify which speech and noise factors have the largest impact on tool-calling accuracy, enabling targeted comparisons across model families.

This paper makes the following primary contributions:

1. We propose **BFCL Audio**, the first benchmark to systematically evaluate *audio* function calling in LLMs under realistic acoustic perturbations and clarification-driven interaction.

2. We curate and release *6.2K* tasks spanning BFCL Text Audio and BFCL True Audio with ground-truth tool-call references and automated grading.

3. We introduce a controllable pipeline for realistic synthetic audio generation and augmentation, enabling fine-grained control over speech factors and background noise conditions.

4. We propose an audio-specific failure-mode taxonomy and conduct a large-scale study that identifies dominant error types and quantifies which speech/noise factors most strongly affect accuracy.

With our comprehensive harness, we hope BFCL Audio will become a standard framework for audio tool-call evaluation.

## 2. Related Work

**Tool-Calling Benchmarks.** A growing body of work benchmarks tool use in text-only settings. API-Bank (Li et al., 2023) and GORILLA (Patil et al., 2023) evaluate mapping written instructions to structured calls. BFCL (Patil et al., 2025) and $\tau$-Bench (Yao et al., 2024; Barres et al., 2025) expand this paradigm to multi-turn, tool-agent-user interaction with intermediate dialogue. These benchmarks have been essential for progress on structured tool invocation, but they typically assume clean, well-formed text inputs and therefore do not surface perception-induced failures that arise when requests are delivered as speech (e.g., ASR errors, homophones, noise, and disfluencies) or when the system must decide whether to ask clarifying questions. Motivated by this gap, prior work in speech has primarily evaluated *perception* robustness, while a smaller set of benchmarks has begun to evaluate tool use in spoken settings. BFCL Audio complements these lines of work by evaluating tool calling under realistic speech conditions and by explicitly supporting clarification-driven interactions.

**Audio Perception Benchmarks.** Speech and audio benchmarks traditionally emphasize perception/recognition robustness and acoustic robustness, rather than downstream tool invocation. This differs from *audio tool use*, where the goal is to map speech to *correct structured actions* (function names and arguments), and where small perception errors can result in significant execution failures. Datasets such as CHiME (Barker et al., 2018) and VOiCES (Richey et al., 2018) study ASR under realistic noise conditions, and resources such as MUSAN (Snyder et al., 2015) provide background noise for augmentation. Classic conversational corpora (e.g., Switchboard (Godfrey et al., 1992) and Fisher (Cieri et al., 2004)) and the "cocktail party" line of work (Cherry, 1953) have similarly driven progress on speech modeling under interference. These benchmarks establish realistic acoustic conditions, but they do not measure whether a system can *execute* the correct tool calls under such conditions. BFCL Audio builds on this line by shifting the evaluation target from word error rate to *correct tool execution*: we test whether perception errors propagate into incorrect function names/arguments and whether models appropriately enter clarification loops.

**Audio Tool-Use Benchmarks.** A small but growing line of recent work evaluates voice assistants and audio-language models in settings that *touch* on tool use. CAVA (Held et al., 2025) provides a broad, product-oriented evaluation suite (e.g., turn-taking, latency, tone awareness) and includes a spoken task-oriented parsing task whose function-calling evaluation focuses on *function-name* matching, rather than argument-level correctness. AU-Harness (Surapaneni et al., 2025) offers an evaluation framework and includes a "speech function calling" task based on BFCL-style(Patil et al., 2025) tool invocation, constructed by converting text instructions into spoken counterparts via TTS. While valuable, these evaluations largely operate at a surface layer: they are primarily single-turn and do not stress the full end-to-end tool-calling pipeline with controlled acoustic perturbations and clarification behavior, nor do they enforce argument-level correctness in the way required by many applications. BFCL Audio advances this line by (i) evaluating both function names and *argument values* with automatic

metrics, (ii) supporting multi-turn clarification loops, and (iii) comprehensively benchmarking the Text-Model-in-the-Loop (ASR→LLM→TTS) pipeline alongside end-to-end audio-in tool calling under matched, controllable speech and noise conditions.

In summary, prior benchmarks primarily evaluate tool use in text-only settings, while speech datasets focus on recognition robustness rather than downstream tool invocation. BFCL Audio bridges this gap by providing an audio-centric tool-calling benchmark with both pipelined and end-to-end evaluation suites, controlled perturbations, and clarification-aware interaction.

## 3. Data Curation

This section details the construction of two complementary datasets, **BFCL True Audio** and **BFCL Text Audio**. BFCL True Audio contains augmented audio utterances, while BFCL Text Audio provides their corresponding ASR transcripts. Together, they enable evaluation of multimodal agents across audio input and audio-derived textual representations.

### 3.1. BFCL True Audio

BFCL True Audio converts the original clean textual queries into realistic spoken utterances through a four–stage process: (1) *natural paraphrasing*, (2) *controllable speech-noise injection*, (3) *synthetic speech generation*, and (4) *real-world acoustic augmentation*. We describe each below.

#### 3.1.1. NATURAL PARAPHRASING:

To create a challenging and diverse set of prompts, we start with text-only function calling queries derived from the BFCL (Patil et al., 2025), including single-turn and multi-turn queries curated by experts and the community. To transform these queries to resemble natural, speech-like language typical of colloquial conversations, we filter out queries that contain special characters or symbols. We then rewrite each user query into a more natural conversational tone. This is critical, as we discuss below; for example:

*Original:* "I need to send a letter to Liam Neeson. Find his contact information."

*MFCL processed text:* "Um, can you get Liam Neeson—that's L-I-A-M N-E-E-S-O-N—Liam Neeson's contact info so I can send him a letter?"

#### 3.1.2. GENERATING SPEECH TRANSCRIPTS:

Spontaneous speech naturally contains numerous disfluencies such as *filled pauses* ("um," "uh"), *word repetitions* ("the the station"), *hesitations*, *elongations*, and *mid-sentence restarts* ("I want— I mean, we should..."). We

broaden this set to include *self-corrections* ("...no, sorry, I meant..."), *false starts* ("Hey, could you—uh, can you..."), *casual contractions* ("gonna," "wanna"), *conversational markers* ("you know," "I mean"), and explicit *spelling or symbol pronunciations* (e.g., "Contact the admin at j–o–h-n at gmail dot com."). Additionally, structural edits such as *preposition dropping* and *sentence restructuring* further mimic everyday speech.

Balancing realism and intelligibility is non-trivial: too many disfluencies can overwhelm short queries, while too few leave them unnaturally pristine. To achieve this, we first extract 23 distinct speech features by analyzing real spontaneous speech and partition them into mutually exclusive classes. The LLM-based controller then selects a subset of features from each class, with the selection dependent on the length of the query and the current noise level of the test case. Once selected, the features are applied sequentially in a cascading fashion, independently modifying the utterance to simulate compounding speech-feature errors. To ensure that critical information is preserved, the controller has access to the full function signature of any downstream tool call and can veto transformations that would alter spellings, remove essential content, or otherwise compromise the correctness of the function-calling output. Fig. 1 provides an overview of the pipeline.

**Ablation-informed weighting.** To ensure BFCL Audio is challenging under realistic failure cases, our controller selects speech features that are *appropriate for the given query*, while biasing selection toward features that have the largest measured impact on downstream tool-use accuracy in controlled ablations. Concretely, we upweight speech features that most frequently induce tool-calling failures in BFCL Text Audio (Appendix C, Fig. 5), and we emphasize acoustic conditions (e.g., competing speech, network impairment) that most degrade intent preservation (Section 6.2).

#### 3.1.3. SYNTHETIC SPEECH GENERATION:

The disfluency-augmented text is rendered into waveform audio using a diverse set of voice clones. Instead of training an in-house TTS architecture, we use a third-party Voice Cloning API, supplying hours of recordings from the Mozilla Common Voice Dataset (Ardila et al., 2020) to create approximately 225 distinct voice clones that vary in speaker identity, accent, prosody, and speech patterns such as pauses and intonation (see Appendix F for full implementation details). This approach captures the natural variability found in real-world speech, including differences in speech style and rhythm, which is critical for evaluating multimodal agents under realistic conditions. By generating audio from the disfluency-augmented text with these high-variance voice clones, we produce samples that are both intelligible and representative of the diversity encountered

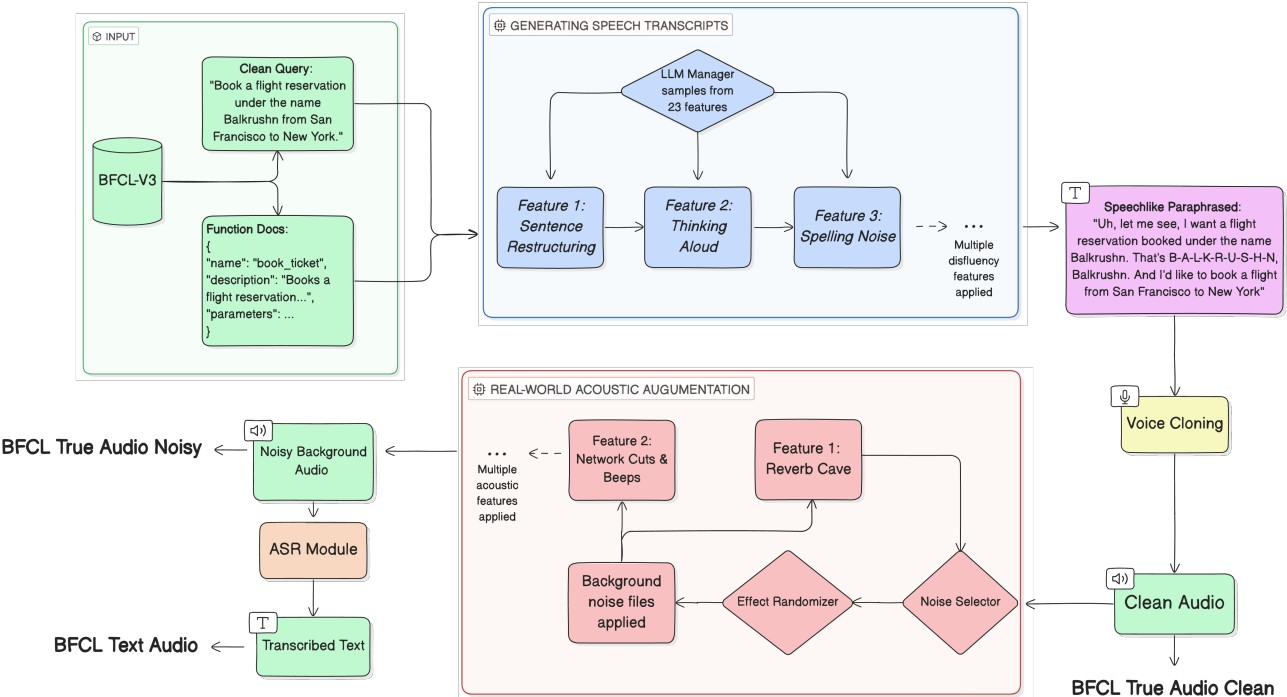

*Figure 1.* Overview of the BFCL-Audio data curation pipeline. Clean text-based function-calling queries are first rewritten into natural, speech-like utterances via controlled disfluency injection, guided by query complexity and tool constraints. The resulting transcripts are rendered into synthetic speech using diverse voice clones to form clean snippets for BFCL True Audio. Real-world acoustic augmentations (e.g., background noise, reverberation, and network artifacts) are then applied creating noisy background samples for BFCL True Audio, and the final audio is then also transcribed by ASR systems to produce BFCL Text Audio for text-based evaluation.

in everyday spoken queries.

### 3.1.4. REAL-WORLD ACOUSTIC AUGMENTATION:

Although voice clone audio output is diverse, it is still somewhat clean. We therefore apply four augmentation modules that emulate the degradations common in real deployments (Fig. 1). **Additive Environmental Noise Background:** audio from the MUSAN corpus(Snyder et al., 2015) and CHiME-5 archive(Barker et al., 2018) is mixed at random signal-to-noise ratios (SNRs). **Competing Speech (Double-Talk):** The target utterance is overlaid with interfering speakers sampled from conversational corpora at varied SNRs, assessing robustness in acoustically dense environments like open-plan offices or public transport. **Network Impairments:** Packet loss, jitter, and truncation are synthesized to stress models operating over VoIP or cellular links. We randomly drop 10–40 ms frames, micro-loop short segments, or truncate the tail of the utterance to emulate latency and test the model's ability to act on incomplete information or seek clarification. **Device and Room Effects:** Room impulse responses add reverberation; non-linear transfer curves simulate clipping; random gain shifts emulate speaker-to-microphone distance changes; and short bursts of rubbing or cable noise model mechanical interference.

The parameters of each module are tuned so that queries remain intelligible to robust systems while exposing brittleness in fragile pipelines.

### 3.2. BFCL Text Audio

BFCL Text Audio builds on top of BFCL True Audio by transcribing each synthetic utterance with three distinct ASR engines, exposing systematic ASR variability across providers (see Appendix F for engine details). Each transcript is stored separately, exposing systematic ASR variability. For example:

· **Generator 1:** "Um, can you get Liam Neeson—that's L-I-A-M N-E-E-S-O-N—Liam Neeson's contact info so I can send him a letter?"

· **Generator 2:** "Um, can you get Liam Neeson, that's L-I-A-M N-E-E-S-O-N, Liam Neeson's contact info so I can send him a letter?"

· **Generator 3:** "Can you get Liam *Neeeson*? That's L-I-A-M N-E-E-S-O-N, Liam *Neeeson*'s contact info so I can send him a letter?"

BFCL Audio exploits these divergences to overcome the brittleness of single-provider ASR, thereby providing a com-

prehensive and robust end-to-end view.

## 3.3. Human Perception Validation

Our goal with the BFCL Audio data curation pipeline is to approximate the range of acoustic and linguistic conditions encountered by deployed voice assistants, rather than to perfectly reproduce any single conversational corpus. Audio realism is critical for two reasons: (i) to ensure that the audio itself is perceptually natural and captures real-world variability for end-to-end models (**BFCL True Audio**), and (ii) to ensure that ASR-based text generation for pipelined models (**BFCL Text Audio**) is performed on audio representative of realistic conditions, to guarantee that the dataset reflects real-world conditions and evaluation of perception-induced errors is meaningful.

To directly verify perceptual naturalness, we conducted a human evaluation on our synthetic audio. We sampled 200 utterances (100 clean, 100 noise-augmented) and recruited 20 raters (internal volunteers). Each rater evaluated a random subset on a 1–5 scale for: Speech Naturalness ("Does the voice sound like a plausible human speaker?"), and Acoustic Realism ("Does the background / channel sound like a plausible real-world condition?"; asked only for noisy clips). As Table 1 shows, in 83% of noisy clips, raters gave a score of 3 or higher on both metrics, indicating that listeners perceive our synthetic audio as natural and realistic.

*Table 1.* Human evaluation of synthetic audio on a 1–5 Likert scale (1 = very unnatural, 3 = acceptable, 5 = highly natural). Both clean and noisy synthetic speech achieve mean naturalness scores well above the acceptability threshold ($\geq 3$), indicating that the generated speech is generally perceived as plausible. Scores are reported as Mean $\pm$ SD. Acoustic realism is only defined for noisy audio, since no acoustic noise augmentations are applied to the clean condition.

| Metric | Clean Audio | Noisy Audio |
|---|---|---|
| Speech Naturalness | $4.1 \pm 0.7$ | $3.9 \pm 0.8$ |
| Acoustic Realism | N/A | $3.8 \pm 0.7$ |

To more directly assess perceptual closeness to real speech, we conducted a matched pairwise CMOS study comparing our clean synthetic audio against real human recordings of the same rewritten transcripts. Twenty raters evaluated 100 matched pairs (2,000 total judgments): ground-truth speech was preferred in 58% of trials, BFCL clean audio in 12%, and 30% were ties or no preference, yielding a mean CMOS of +0.68 in favor of ground truth. While this confirms that our synthetic audio is distinguishable from real recordings, our goal is not imperceptibility but perceptual realism sufficient for benchmarking audio function calling, a bar this evaluation supports.

## 4. Evaluation Setup

**BFCL Text Audio** evaluates (i) correct function invocation and (ii) robustness to audio artifacts. **Turn semantics:** Unlike existing text-based benchmarks, where non-tool messages end a turn, spoken interactions often require confirming spellings or values due to homophones and ASR errors. To avoid penalizing necessary clarifications, we allow whitelisted confirmation turns (validated against a dictionary) without ending the interaction. A system prompt informs the model of the audio setting (Appendix E). **Clarification mechanism:** An LLM judge validates clarification requests; if valid, a simulated user provides a brief, whitelisted reply (e.g., spelling of proper nouns). (Appendix D). **Scoring (single-turn):** We parse the predicted tool call into an AST and score it by function-name match and argument-level correctness. Missing required parameters or extra arguments are incorrect; values are compared with type-aware normalization (e.g., strings are case/whitespace-insensitive) and list order is enforced unless specified otherwise. Optional parameters may be omitted when defaults exist. **Scoring (multi-turn):** A multi-turn interaction is correct iff *every* turn passes both (i) *state-based* matching of the post-execution system state to the labeled state and (ii) *response-based* verification that the model executed the minimal required tool-call path (to avoid "guessing" on read-only steps). Clarification turns are permitted but not scored.

**BFCL True Audio** Same as BFCL Text Audio, but with simulated synthetic audio replies for whitelisted clarifications.

## 5. Analysis

In this section, we analyze model failure modes under both speech disfluencies and noise injections. Table 2 reports progressively harder tool-calling structures. The single-turn block reports *Simple* = one valid tool call; *Multiple* = choose among several tools; *Parallel* = repeated calls to the same tool; *Parallel Multiple* = multiple tools with repeated calls. It also separates *Expert Curated* examples, which systematically cover functional patterns, from *Community Sourced* examples, which are drawn from real-user-style inputs and are therefore noisier. The multi-turn block reports *Base* = standard sequences, *Miss Func* = no valid function exists, and *Miss Param* = missing arguments that should trigger clarification. *Relevance/Irrelevance* measure whether the model calls a tool only when warranted. Audio-specific errors affect different parts of the pipeline differently, including tool selection, argument grounding, call composition, and decisions around abstention or clarification; accordingly, we break down function-calling errors into these categories for finer-grained insight into model behavior along with more precision in error characterization.

## 5.1. Failure Modes

Switching from native text inputs to audio-based interaction introduces a systematic drop in function-calling accuracy across models, reflecting a fundamental modality gap. Appendix G (Fig. 8, 9, 10) show this gap in depth, across model families, motivating detailed failure-mode analysis.

Background noise sharply degrades every end-to-end (E2E) model we tested (Table 2). Even the strongest system, GPT-4o-audio, loses **9.5 %**. Under noise, the dominant error source shifts from *detecting a request* to *transcribing its details*, producing subtle yet harmful *semantic errors*: the assistant executes an action, but the wrong one.

We observe six recurrent failure modes (FMs):

**FM 1: Intent Blending** Background speech merges with the user's command (e.g., a coworker says "cancel the meeting," which contaminates a flight-booking request).

**FM 2: Premature Execution** A sudden loud sound (door slam, siren) is misheard as end-of-utterance, prompting action on an incomplete command.

**FM 3: Parameter Distortion** Microphone artifacts (clipping, rubbing) mutate key tokens; "fifty" becomes "fifteen," yielding a valid but incorrect function call.

**FM 4: Clarification Misfires** The model asks irrelevant clarification questions that are not about named entities, especially in pipelined models with noisy ASR (Fig. 3). ASR transcriptions add noise, creating more uncertainty and leading to more irrelevant clarification requests. End-to-end (E2E) models perform better due to lower uncertainty from integrated audio-language understanding.

**FM 5: Conversational Drift** Instead of emitting the required function call, the model slips into a conversational response. RLHF-tuned E2E models default to helpful dialog when uncertainty spikes, trading schema compliance for user-friendly chatter. (Fig. 3)

**FM 6: Named-Entity Errors** Mistranscribed names, places, or addresses derail argument generation, and the model never asks for confirmation, instead generating a faulty call (Fig. 3). Noisy audio worsens this across all architectures, but is most acute for noisy E2E runs. For pipelined models in particular, the ASR stage distorts key tokens while still producing grammatically plausible text, leading to error propagation, as the following example shows.

· *Original Query:* "Can you book a flight to Austin for tomorrow morning?"

· *Audio Condition:* True audio with medium-density cafe background noise.

· *ASR Transcript (Clean Audio):* "Can you book a flight to **Austin** for tomorrow morning?"

· *ASR Transcript (Noisy Audio):* "Can you book a flight to **Boston** for tomorrow morning?"

· *Resulting Action:* LLM correctly identified the intent to `book_flight` but received the incorrect entity (`city=Boston`), resulting in a task failure.

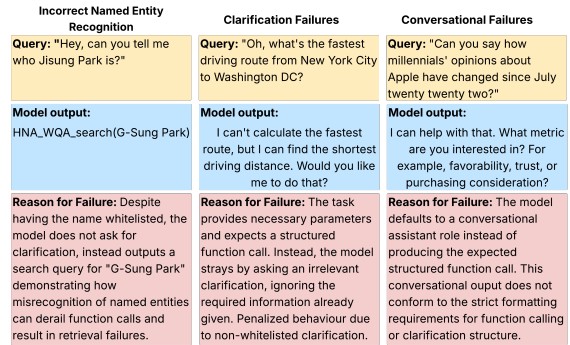

*Figure 2.* Error analysis examples illustrating common function-calling failure modes: named entity misrecognition, unnecessary clarification, and conversational drift. These cases correspond to FM4–FM6 discussed in Section 5.1 and show that failures arise from intent misinterpretation and format non-compliance.

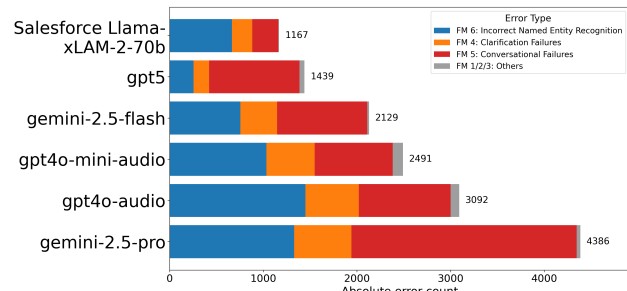

*Figure 3.* Distribution of failure modes across models under BFCL True Audio and BFCL Text Audio. Across architectures, most audio-driven failures stem from conversational drift (FM5) and clarification misfires (FM4), reflecting a broader tendency of models to prioritize safe, dialog-style recovery when confidence drops. Named-entity errors (FM6) appear consistently across both modalities, showing the persistent challenge of robust entity grounding under noisy or ambiguous input. In contrast, intent-level issues (FM1–FM3) remain comparatively rarer, indicating that modern E2E systems correctly capture the task structure but falter on semantic precision and schema compliance under disfluent or noisy speech. See Section 5.1 for specific definitions.

## 6. Ablation Studies

We isolate which linguistic and acoustic factors most degrade audio function calling by intervening at two points in the BFCL Audio pipeline: (i) transcript-level speech features injected during *Generating Speech Transcripts* (Section 3.1) and (ii) the *Real-World Acoustic Augmentation* stage that produces noisy audio (Section 3.1). These controlled interventions complement the aggregate failure-

| Model | Overall | Expert Curated | | | | Community Sourced | | | | Multi-Turn | | | Hallucination Measure | |
|---|---|---|---|---|---|---|---|---|---|---|---|---|---|---|
| | | Simple | Multiple | Parallel | Parallel Multiple | Simple | Multiple | Parallel | Parallel Multiple | Base | Miss Func | Miss Param | Relevance | Irrelevance |
| GPT-4o-audio (Clean Audio) | 60.4 ± 7.1 | 58.6 ± 4.8 | 86.9 ± 5.0 | 80.5 ± 5.9 | 76.0 ± 6.0 | 61.3 ± 6.3 | 67.4 ± 2.8 | 64.3 ± 28.0 | 45.5 ± 21.2 | 37.5 ± 6.5 | 47.5 ± 7.0 | 37.5 ± 6.5 | 72.2 ± 25.1 | 85.4 ± 2.2 |
| GPT-4o-audio (Text) | 58.6 ± 6.8 | 51.1 ± 4.8 | 87.4 ± 4.6 | 77.5 ± 5.9 | 74.5 ± 6.0 | 58.4 ± 6.3 | 66.4 ± 2.9 | 50.0 ± 28.6 | 50.0 ± 22.7 | 38.0 ± 6.5 | 44.0 ± 7.0 | 34.5 ± 6.5 | 77.8 ± 22.2 | 84.1 ± 2.2 |
| Qwen3-Omni-Flash (Clean Audio) | 55.8 ± 7.0 | 58.3 ± 4.8 | 84.9 ± 5.3 | 77.5 ± 6.0 | 74.0 ± 6.3 | 60.7 ± 6.2 | 58.5 ± 2.9 | 53.6 ± 31.0 | 50.1 ± 21.4 | 26.5 ± 5.8 | 33.3 ± 6.3 | 25.5 ± 5.8 | 61.1 ± 23.7 | 87.5 ± 2.1 |
| GLM-4.5 (Text) | 55.2 ± 7.0 | 47.6 ± 5.0 | 87.9 ± 4.5 | 77.5 ± 5.9 | 66.5 ± 6.5 | 60.9 ± 6.4 | 66.4 ± 2.9 | 71.4 ± 25.2 | 59.1 ± 21.9 | 27.5 ± 6.0 | 22.5 ± 6.0 | 25.5 ± 6.0 | 61.1 ± 25.8 | 86.7 ± 2.1 |
| Qwen3-Omni-Flash (Text) | 55.1 ± 6.9 | 52.4 ± 4.9 | 84.2 ± 5.1 | 74.3 ± 6.2 | 72.5 ± 6.3 | 60.7 ± 6.2 | 60.4 ± 3.0 | 53.6 ± 27.5 | 56.8 ± 21.7 | 25.8 ± 5.8 | 31.8 ± 6.3 | 24.0 ± 5.8 | 66.7 ± 24.5 | 87.6 ± 2.0 |
| Claude-Opus-4.1 (Text) | 53.5 ± 7.0 | 54.5 ± 4.9 | 90.0 ± 4.5 | 73.0 ± 6.0 | 60.5 ± 7.0 | 63.0 ± 6.0 | 69.3 ± 2.8 | 78.6 ± 21.4 | 50.0 ± 22.7 | 25.0 ± 6.0 | 17.5 ± 5.5 | 17.0 ± 5.5 | 61.1 ± 25.8 | 81.8 ± 2.4 |
| Gemini-2.5-Flash (Clean Audio) | 53.0 ± 7.0 | 56.8 ± 4.9 | 83.4 ± 5.5 | 77.0 ± 6.0 | 64.5 ± 6.5 | 61.3 ± 6.3 | 63.8 ± 2.9 | 71.4 ± 25.2 | 50.0 ± 22.7 | 16.0 ± 5.0 | 12.5 ± 4.5 | 13.0 ± 4.5 | 66.7 ± 21.5 | 91.5 ± 1.7 |
| Gemini-2.5-Pro (Text) | 51.5 ± 7.0 | 53.6 ± 5.0 | 80.9 ± 5.5 | 71.0 ± 6.5 | 70.5 ± 6.5 | 63.0 ± 6.0 | 54.3 ± 3.0 | 57.1 ± 26.3 | 63.6 ± 20.7 | 13.5 ± 5.0 | 19.5 ± 5.5 | 13.5 ± 5.0 | 55.6 ± 26.8 | 91.1 ± 1.7 |
| xLAM-2-70b-fc-r (Text) | 51.2 ± 7.0 | 54.1 ± 5.0 | 87.4 ± 4.6 | 75.5 ± 6.0 | 59.5 ± 7.0 | 48.6 ± 6.4 | 52.8 ± 3.1 | 42.9 ± 34.0 | 40.9 ± 21.9 | 28.0 ± 6.0 | 28.0 ± 6.0 | 23.5 ± 6.0 | 77.8 ± 22.2 | 80.1 ± 2.4 |
| Gemini-2.5-Pro (Clean Audio) | 51.1 ± 6.9 | 57.9 ± 4.8 | 82.9 ± 5.5 | 74.5 ± 6.0 | 72.0 ± 6.0 | 60.1 ± 6.0 | 49.5 ± 3.0 | 42.9 ± 34.0 | 54.6 ± 21.6 | 15.5 ± 5.0 | 19.0 ± 5.5 | 13.5 ± 5.0 | 50.0 ± 22.2 | 89.6 ± 1.9 |
| GPT-4o-audio (Noisy Audio) | 50.9 ± 7.2 | 43.1 ± 5.0 | 72.9 ± 6.6 | 62.5 ± 6.5 | 62.5 ± 6.5 | 46.5 ± 6.4 | 47.7 ± 3.0 | 35.7 ± 28.0 | 31.8 ± 22.3 | 28.5 ± 6.4 | 39.5 ± 7.0 | 35.0 ± 6.5 | 50.0 ± 22.2 | 85.7 ± 2.2 |
| Grok-4-0709 (Text) | 50.2 ± 7.1 | 53.8 ± 4.9 | 83.4 ± 5.5 | 76.0 ± 6.0 | 67.5 ± 6.9 | 58.9 ± 6.4 | 66.5 ± 2.9 | 71.4 ± 25.2 | 59.1 ± 21.9 | 8.0 ± 4.0 | 3.5 ± 2.5 | 9.0 ± 4.0 | 72.2 ± 25.1 | 84.1 ± 2.2 |
| Qwen3-235B-A22B-Instruct (Text) | 50.1 ± 7.2 | 46.6 ± 5.0 | 86.4 ± 5.0 | 74.0 ± 6.0 | 68.0 ± 6.9 | 53.9 ± 6.4 | 60.9 ± 3.0 | 64.3 ± 28.0 | 59.1 ± 21.9 | 12.0 ± 4.5 | 8.5 ± 4.0 | 7.5 ± 4.0 | 72.2 ± 25.1 | 88.4 ± 2.0 |
| Qwen3-Omni-Flash (Noisy Audio) | 48.2 ± 7.2 | 42.1 ± 5.0 | 71.9 ± 6.6 | 60.0 ± 6.9 | 62.5 ± 6.5 | 48.4 ± 6.4 | 43.8 ± 3.0 | 35.7 ± 28.0 | 40.9 ± 22.5 | 21.0 ± 5.7 | 31.3 ± 6.5 | 26.0 ± 6.0 | 50.0 ± 22.2 | 86.1 ± 2.2 |
| GPT-4o-mini-audio (Clean Audio) | 47.2 ± 7.1 | 51.6 ± 4.8 | 80.4 ± 5.5 | 74.0 ± 6.0 | 67.0 ± 6.9 | 58.4 ± 6.3 | 63.4 ± 2.9 | 71.4 ± 25.2 | 36.4 ± 20.7 | 1.5 ± 2.0 | 1.0 ± 1.5 | 2.0 ± 2.0 | 77.8 ± 22.2 | 82.4 ± 2.3 |
| Gemini-2.5-Flash (Noisy Audio) | 45.6 ± 6.9 | 45.4 ± 5.0 | 72.4 ± 6.6 | 57.0 ± 7.3 | 53.5 ± 7.0 | 49.8 ± 6.4 | 46.4 ± 3.0 | 57.1 ± 26.3 | 50.0 ± 22.7 | 15.5 ± 5.0 | 11.5 ± 4.5 | 9.5 ± 4.5 | 33.3 ± 21.5 | 91.0 ± 1.8 |
| Gemini-2.5-Pro (Noisy Audio) | 45.4 ± 7.2 | 41.0 ± 5.0 | 70.9 ± 6.6 | 57.5 ± 7.2 | 62.5 ± 6.5 | 50.2 ± 6.4 | 39.8 ± 2.9 | 35.7 ± 28.0 | 50.0 ± 22.7 | 13.5 ± 5.0 | 23.0 ± 6.0 | 17.0 ± 5.5 | 50.0 ± 22.2 | 86.4 ± 2.1 |
| GPT-5 (Text) | 41.0 ± 7.0 | 32.1 ± 4.8 | 53.3 ± 7.1 | 55.0 ± 6.8 | 48.5 ± 7.0 | 36.2 ± 6.4 | 39.5 ± 2.9 | 50.0 ± 28.6 | 45.5 ± 21.2 | 11.0 ± 4.5 | 6.5 ± 3.5 | 5.5 ± 3.5 | 44.4 ± 26.8 | 95.0 ± 1.3 |
| GPT-4o-mini-audio (Noisy Audio) | 40.3 ± 6.6 | 36.5 ± 4.7 | 66.3 ± 6.6 | 53.5 ± 7.0 | 52.0 ± 7.0 | 42.0 ± 6.3 | 42.3 ± 3.0 | 50.0 ± 28.6 | 22.7 ± 17.9 | 10.5 ± 4.5 | 3.0 ± 2.5 | 4.0 ± 3.0 | 66.7 ± 21.5 | 84.1 ± 2.2 |

*Table 2.* Model performance on BFCL Text Audio and BFCL True Audio. Evaluation settings: Text = transcribed input, Clean Audio = speech without background noise, Noisy Audio = speech with background noise. End-to-end models on clean audio typically surpass pipelined text-only systems by leveraging contextual and prosodic cues, but their advantage is brittle: noise induces sharp degradation, often below the steadier pipelined baselines, highlighting a trade-off between peak accuracy and robustness.

mode analysis in Section 5 by identifying which perturbations are most harmful *in isolation*.

### 6.1. Impact of Speech Disfluencies

Spoken requests include disfluencies (filled pauses, repetitions, self-corrections, restarts) and discourse phenomena (ellipsis, proforms) that can under-specify arguments. BFCL Audio's transcript generator (Section 3.1) lets us inject such features while constraining edits to preserve the underlying tool-call semantics. Here we apply one feature at a time to identify which linguistic perturbations most reliably induce downstream tool-use failures.

**Setup.** We apply one speech feature at a time and exclude queries that already fail in the unperturbed condition, isolating the causal impact of each feature. Full counts and filtering details are provided in Appendix C.

We evaluate both deployment styles supported by BFCL Audio: BFCL Text Audio (pipelined ASR→LLM→tools) and BFCL True Audio (end-to-end audio-in→tool calls). For BFCL True Audio, we use *clean* audio (no acoustic augmentation) to isolate linguistic effects.

#### 6.1.1. BFCL TEXT AUDIO: ASR BOTTLENECKS

For each feature-perturbed utterance, we synthesize audio, transcribe it with three ASR engines, and then evaluate function-calling accuracy with text-only function-calling models. Fig. 5 (Appendix C) summarizes the total downstream failures attributable to each injected speech feature.

**Discourse-altering features dominate failures.** The most disruptive features are self-corrections, ellipsis/proforms, sentence restructuring, restarts/repairs, and detail dropping (Fig. 5). These changes alter discourse structure or semantic specificity (not just surface form), and therefore directly interfere with intent preservation and argument grounding.

**Self-corrections and restarts trigger clarification and drift.** Mid-utterance revisions are often transcribed verbatim, forcing the model to interpret both the original and corrected content. This increases uncertainty and yields **FM4 (Clarification Misfires)** and **FM5 (Conversational Drift)**. See Appendix C.2 for an illustrative example.

**Ellipsis, proforms, and detail dropping break argument grounding.** Vague references (e.g., "do that") and omitted qualifiers remove explicit spans needed for parameter binding. Models then guess missing values or act on partial information, leading to **FM2 (Premature Execution)** and **FM6 (Named-Entity Errors)**. See Appendix C.2 for an illustrative example.

**Sentence restructuring shifts intent and parameters.** Reordering or paraphrasing can subtly change how models map utterances to functions, sometimes selecting related but incorrect tools or introducing unsupported arguments, consistent with **FM1 (Intent Blending)** and **FM3 (Parameter Distortion)**. See Appendix C.2 for an illustrative example.

**Surface-level disfluencies are largely benign.** Fillers, repetitions, casual pronouns, contractions, and emotional markers generally preserve semantic content and argument structure, allowing ASR and downstream models to recover intent reliably. Across all three ASR engines, discourse-altering features account for the majority of downstream failures; the full per-provider breakdown is shown in Appendix C (Fig. 6).

#### 6.1.2. BFCL TRUE AUDIO: DIRECT AUDIO BOTTLENECKS

We next evaluate BFCL True Audio, where audio-capable models produce tool calls directly from speech. Fig. 7 (Appendix C) reports the corresponding feature-level error counts.

In contrast to the ASR-mediated setting, errors in the end-to-end audio models are distributed much more uniformly across speech features. Most disfluencies and surface-level variations result in similarly low failure rates, with *filepath verbalization* standing out as the sole feature that consistently produces a substantially higher number of errors.

This pattern likely reflects the difficulty of grounding long, symbol-heavy sequences when they are spoken aloud. File paths contain slashes, dots, and mixed alphanumeric tokens that do not map cleanly to natural speech, creating a mismatch with the discrete string arguments required by function schemas.

**Takeaway.** Speech-feature brittleness depends on deployment style: ASR-mediated pipelines fail most often under discourse-altering/argument-underspecifying edits, while end-to-end audio models are comparatively robust to these edits but struggle when exact symbolic reconstruction is required (e.g., file paths).

## 6.2. Impact of Acoustic Noise

We ablate the acoustic augmentation modules used in BFCL Audio's *Real-World Acoustic Augmentation* stage (Section 3.1). Starting from clean utterances, we apply multiple noise/effect types to generate noisy variants and study how different acoustic conditions corrupt the *function-critical* content needed for correct tool calls. Full counts are provided in Appendix C.

**Metrics.** We evaluate noise via transcript *intent viability*: **intent preservation** (binary) and **intent confidence** (0–1). We additionally bucket failures into coarse diagnostic types (e.g., entity confusion, deletion, semantic drift) to connect noise to the failure modes in Section 5, and report WER only as a sanity check.

**Noise conditions.** We evaluate environmental noise (MUSAN/CHiME-5 mixes; fades/gain changes), spatial effects (reverb echo/cave), network impairments (packet loss/truncation with 10–40 ms drops), device artifacts (mic rubbing/mechanical interference), and competing speech.

### 6.2.1. RESULTS

Table 3 summarizes the impact of each noise type, ranked by intent preservation rate.

**Competing speech dominates.** Overlapping speakers are most harmful because interfering words are transcribed as plausible in-domain content and contaminate function arguments, driving **FM1 (Intent Blending)**.

**Reverb and network cuts are next, with distinct signatures.** Reverb increases phonetic confusions that corrupt entities, while packet loss/truncation deletes spans and increases **FM2 (Premature Execution)**.

*Table 3.* Impact of acoustic noise types on function-calling viability. Intent preservation and confidence are the primary metrics; WER is included as a supplementary sanity check only, as it does not reliably track tool-use failures; mic rubbing, for example, shows near-zero intent loss despite non-trivial WER, whereas competing speech causes severe intent loss at comparable WER. *Samples Evaluated* denotes utterances per condition; larger counts for background noise reflect multiple noise types and intensities, while isolated conditions have smaller sets. Comparisons remain valid despite unequal group sizes as metrics are computed per condition.

| Noise Type | Intent Pres. ↑ | Confidence ↑ | Samples evaluated |
|---|---|---|---|
| *Severe Impact (Intent Pres. < 60%)* | | | |
| Competing speech | 42.4% | 0.578 | 243 |
| Reverb echo | 48.8% | 0.582 | 162 |
| Network cuts | 50.6% | 0.641 | 162 |
| *Moderate Impact (60% ≤ Intent Pres. < 90%)* | | | |
| Background noise (all) | 74.6% | 0.806 | 5346 |
| Reverb cave | 79.0% | 0.827 | 81 |
| *Minimal Impact (Intent Pres. ≥ 90%)* | | | |
| Volume fluctuation | 96.3% | 0.940 | 162 |
| Audio fade | 98.8% | 0.988 | 162 |
| Mic rubbing | 98.8% | 0.987 | 81 |
| Mumbling effect | 100.0% | 0.991 | 81 |

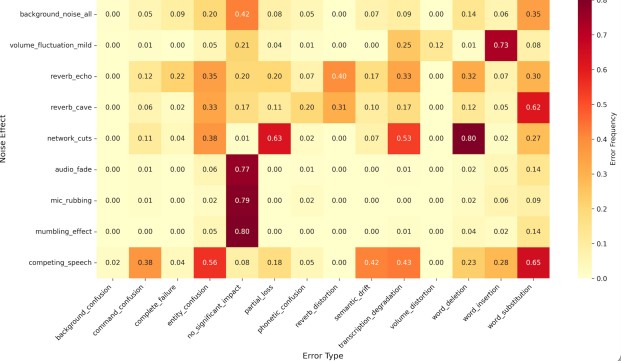

*Figure 4.* Error type distribution across noise conditions. Competing speech and reverb show high rates of entity confusion; network cuts predominantly cause word deletion.

**Device artifacts are mostly benign.** Microphone rubbing, mumbling, fades, and gain changes often preserve function-critical content despite noticeable transcript differences, illustrating why WER alone can be misleading for tool use.

**Error-type breakdown.** Fig. 4 shows distinct signatures: competing speech and reverb exhibit high entity confusion (driving FM6), network cuts are dominated by word deletion/partial loss, and competing speech also induces semantic drift (grammatical but wrong intent).

**Intent-category breakdown.** Under competing speech (the hardest condition), dictation tasks with dense slot values degrade the most, and navigation exhibits the largest gap relative to the clean condition (Table 4).

**Takeaway.** The critical acoustic bottlenecks are the "cocktail party" setting (interference that induces plausible-but-wrong arguments) and network impairments (span deletion), suggesting robustness efforts should prioritize interference

*Table 4.* Intent preservation by utterance category under competing speech.

| Intent Type | Overall | + Competing Speech |
|---|---|---|
| Navigation | 77.3% | 25.0% |
| Dictation | 72.4% | 33.3% |
| Command | 74.1% | 40.5% |
| Question | 74.8% | 47.4% |

handling and uncertainty-aware clarification over device-noise modeling.

## 7. Limitations

BFCL Audio is designed to approximate deployment-relevant speech and acoustic conditions in a controlled and reproducible way; nevertheless, it remains a synthetic benchmark. Voice-cloned speech, rewritten transcripts, and simulated acoustic perturbations cannot fully capture the spontaneity, recording variability, and environmental complexity of naturally occurring interactions. Benchmarks built from real recordings, manual transcripts, and authentic deployment settings, such as office environments with naturally occurring background noise, would provide stronger ecological validity. We therefore view BFCL Audio as a controlled diagnostic benchmark that complements, rather than replaces, evaluation on fully real-world data, and we consider the construction of such naturalistic corpora an important direction for future work.

## 8. Conclusion

We introduced BFCL Audio, a benchmark for evaluating end-to-end audio function calling under realistic speech and acoustic conditions. BFCL Audio contains 6.2K expert-verified tasks across two suites that mirror common deployments: BFCL Text Audio for ASR-mediated pipelines and BFCL True Audio for direct audio-in tool use. By pairing a controllable generation/augmentation pipeline with automatic, judge-free grading, BFCL Audio enables scalable and diagnostic evaluation of both function selection and argument fidelity. Across a broad set of models, we found that background noise, especially competing speech, and discourse-altering disfluencies substantially degrade tool-calling reliability, with failures driven by named-entity errors, parameter distortion, and clarification misfires. These results underscore that reliable speech agents require robustness beyond text function calling, including uncertainty-aware confirmation strategies, improved handling of interference and underspecified spoken requests. We release BFCL Audio, the evaluation harness, and the audio pipeline to support future work on speech-based tool use.

# Impact Statement

This paper introduces BFCL Audio, a benchmark and evaluation harness for end-to-end audio function calling. By providing automatic, diagnostic evaluation under realistic speech and acoustic conditions (e.g., accent variation, disfluencies, and background noise), our work can help researchers and practitioners build more reliable and accessible speech-based agents, and reduce user harm caused by silent transcription or argument errors when tools are executed. At the same time, improving robustness of voice-to-tool systems could lower the barrier to deploying audio agents in harmful settings (e.g., surveillance, coercive monitoring, or scalable social engineering), and model performance may remain uneven across accents and acoustic environments, creating disparate reliability across user groups. We mitigate these risks by releasing BFCL Audio primarily as an evaluation resource with clear documentation and intended-use guidance, by reporting condition-specific failures to support auditing, and by including licensing restrictions described in our Ethics Statement to discourage malicious use.

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

# A. Ethics Statement

Our work introduces BFCL Audio, an audio function-calling evaluation that combines publicly available text and audio data. All source datasets are either (i) released under permissive licenses (e.g., CC-BY, CC-BY-SA) or (ii) in the public domain; we redistribute only metadata and references, never raw copyright-protected content. No human-subject experiments were conducted, and no personally identifiable information is included, so institutional review board (IRB) approval was not required. We audited BFCL Audio for sensitive attributes (gender, race, religion) and found no representation of sensitive categories; nevertheless, downstream users should be aware that biased function schemas could amplify demographic stereotypes. To mitigate misuse, we provide detailed documentation describing collection, filtering, and annotation rules; a usage license that forbids deploying BFCL Audio to train or evaluate models intended for surveillance, disinformation, or other malicious purposes; and release a standardized bias-and-toxicity evaluation script to help practitioners quantify harmful behaviors. The authors declare no conflicts of interest or third-party sponsorship that could unduly influence the results.

# B. Reproducibility Statement

We have taken several steps to facilitate reproducibility: All preprocessing code, function schemas, and evaluation scripts are available publicly online in an anonymous repository. Section 3 details dataset construction. Section 4 describes the metrics.

# C. Additional Ablation Results and Examples

**Disfluency ablation setup details.** We sample 105 base BFCL queries and apply a single speech feature per query, yielding 2,520 feature-perturbed instances (24 transcript-level features applicable under our single-feature constraints).[1] To isolate the causal effect of feature injection, we exclude 21 base queries that already fail in the unperturbed condition.

**Noise ablation setup details.** Starting from 243 clean utterances, we apply nine noise/effect types to generate 6,480 noisy variants.

## C.1. Ablation-informed feature impact

Figure 5 summarizes the total number of downstream function-calling failures attributable to each injected speech feature in BFCL Text Audio. Figure 7 reports the corresponding end-to-end audio feature-level error counts under clean audio.

## C.2. Qualitative examples for speech-feature ablations

**Self-corrections and restarts (FM4/FM5):**

| |
|---|
| **Content:** "weather in India using search web tool" |
| **Transcript:** "weather in India using search... no, using the search web tool" |
| **Model Result:** "Could you please clarify your query? Are you looking for the weather in India, or is there something else you would like to search for related to India?" |
| **Error:** Irrelevant Clarification |

**Ellipsis/proforms and detail dropping (FM2/FM6):**

| |
|---|
| **Content:** "Could you help me retrieve the list of exports for my bot using the identifier 'my-bot-id'..." |
| **Transcript:** "Could you help me do that thing where you get the list of exports for my bot using the identifier 'my-bot-id'..." |
| **Model Result:** "Of course, I can help with that. I will use the bot id you provided, my-bot-id, and set the version to v2...Do you want to proceed?" |
| **Error:** Conversational Drift |

**Sentence restructuring and schema-sensitive formats:**

---

[1]BFCL Audio defines 26 total speech features in the generator (Section 3.1); two features are excluded here because they are not meaningfully applicable as *single-feature* interventions without changing task constraints.

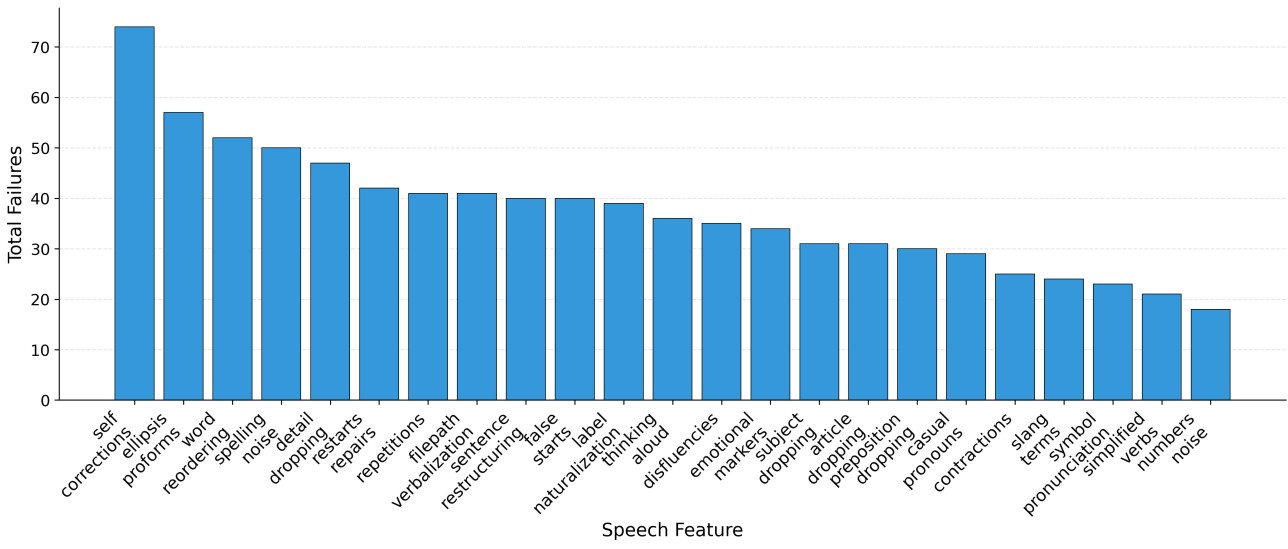

*Figure 5.* Comparison of ASR-induced failure distributions across speech features in bfclTextAudio. The figure shows how specific linguistic and prosodic features amplify transcription errors that propagate to downstream function-calling. Failures concentrate around structural features like self-corrections, ellipsis, and word reordering, indicating high ASR sensitivity to discourse disruption. In contrast, surface features such as fillers and slang have limited downstream impact, underscoring that ASR brittleness is shaped more by semantic restructuring than phonetic noise.

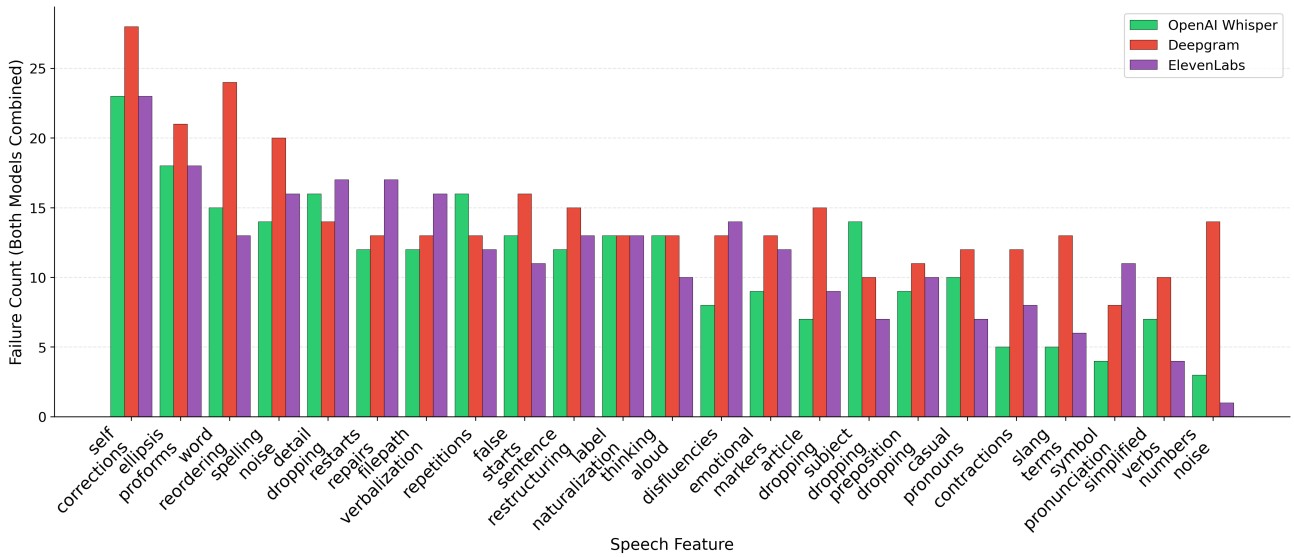

*Figure 6.* Downstream function-calling failures attributable to each speech feature, stratified by ASR provider.

---

**Content:** "Could you tell me the weather forecast for Marshall, MN on March 5th 2023?"

**Transcript:** "What's the weather forecast for Marshall, Minnesota on March 5th, 2023?"

**Error:** ASR expands an abbreviation (MN→Minnesota), which violates a schema constraint that expects a specific string format.

---

## D. Prompt to the LLM model for judging clarification

The LLM model sees the intended request, the ASR text, the assistant's message, and the allowed clarification keys. It approves only if the assistant is explicitly confirming spellings/values that appear in the whitelist. Otherwise, it rejects.

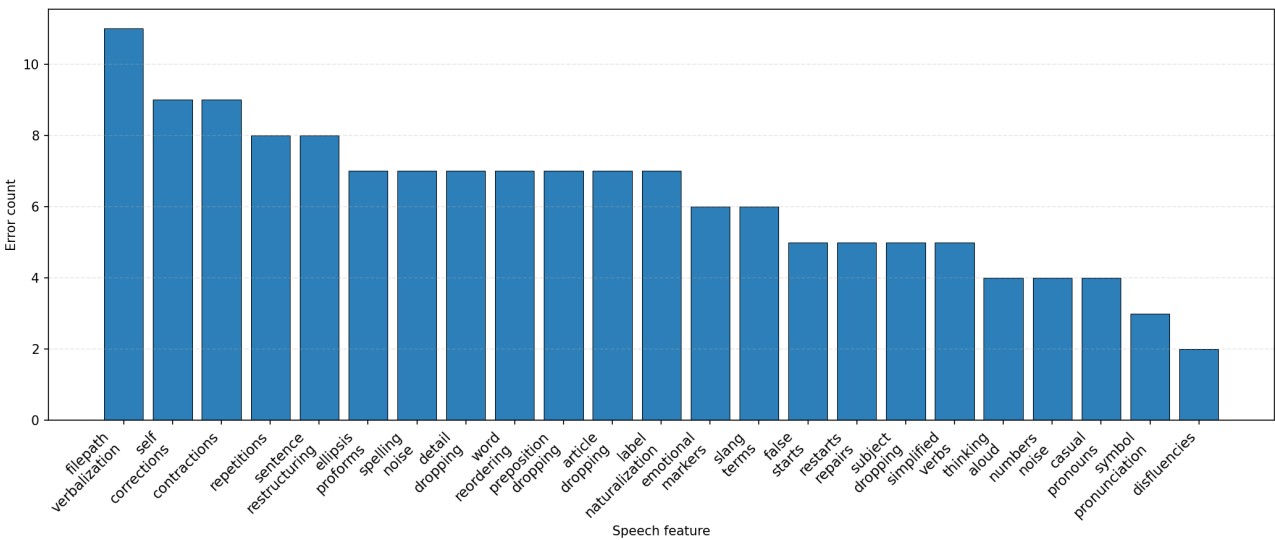

*Figure 7.* Error counts for end-to-end audio function calling across speech features using BFCL True Audio with clean audio inputs. Errors are largely uniform across features, with filepath verbalization emerging as the dominant source of failures.

```
You are a judge for an audio-chat scenario where a user speaks and an ASR system
    transcribes their speech for the assistant. The assistant only sees text (the ASR
     transcript), which is likely to contain transcription errors.

You are given:
- intended_request: the user's original, ground-truth intent.
- asr_text: the ASR-transcribed text the assistant saw.
- allowed_clarifications: a set of fields with canonical spellings/values the user
    can confirm (e.g., names, IDs, emails, dates, numbers).
- assistant_message: the assistant's latest message.

Your job: decide whether assistant_message is a clarifying question specifically
    about spelling/verification of intent or exact strings/values that could
    plausibly be misheard (e.g., names, organizations, emails, serials/IDs, numbers,
    dates, addresses, SKUs). Do not allow general follow-ups (preference, steps to
    proceed, etc.).

Decision rules:
1. Classify the message as a spelling confirmation only if it explicitly asks to
    verify the exact spelling/format/value of one or more items (e.g., "Is it Mikaela
     or Michaela?", "Can you spell the email?", "Is the order number A1B-52?").
2. The request must be reasonable given the ASR risk (i.e., the item is a proper
    noun, key value, or easily misheard token relevant to the task).
3. To approve (allowed=true), all the topics the assistant asks to confirm must be
    present in allowed_clarifications. If any requested item is absent or ambiguous,
    set allowed=false.
4. Output only a JSON object with two fields:
- allowed: boolean
- message: string (a concise simulated user reply only when allowed=true; otherwise
    empty "").
1. When allowed=true, compose message by supplying only the requested values with
    correct spelling/format from allowed_clarifications. Keep it brief (one short
    sentence or a compact list). Do not include extra commentary, JSON, or fields the
     assistant didn't request.
2. If the assistant's message is not a confirmation request, touches topics outside
    spelling/format/intent verification, or requests values not available in
```

```
      allowed_clarifications, return allowed=false with message="".

Edge cases:
- If the assistant mixes spelling confirmation with unrelated questions, treat it as
     not allowed unless the spelling part stands alone and you can fully answer it
     from allowed_clarifications.
- Treat homophones and near-matches as spelling checks (e.g., "Brian/Bryan", "Steven/
     Stephen", letters vs. digits).
- Normalize case/diacritics but preserve canonical spelling in the final answer.
- Never reveal intended_request verbatim; only return the specific confirmed values.

The user's original intended request is: {the original text mode bfcl question}

The ASR-transcribed output is: {the transcribed text from the audio, which is also
     the input to the model}

assistant_message: {the model's response}

allowed_clarifications (topic -> answer): {the allowed_clarifications}
```

## E. System Prompt for BFCL Text Audio

To inform models that they are in an audio setting, we prepend a short **system prompt** to each conversation:

```
You are a voice assistant that interacts with the user exclusively through spoken
     conversation. You receive user utterances as text transcribed by an upstream ASR
     system and your replies are delivered to the user through a TTS system. Follow
     the rules below at all times:

1. Language

* Mirror the user's language. Respond in the same language detected in the
     transcription.

2. Robustness to ASR Errors (Important)

* Although the upstream ASR system is designed to be robust, it may still make
     mistakes.
* Do not trust the transcription text blindly, especially on important information.
     You should assume the transcript may contain recognition mistakes.
* If the text appears garbled, double-check with the user instead of guessing.

3. Clarity for TTS

* When responding to the user, you should **spell out acronyms** as separate letters
     with spaces ("A I M L"), and **chunk long numbers** into 2- or 3-digit groups,
     separated by short pauses ("one-two-three, four-five-six").
* Favor spoken-language style: short sentences, everyday vocabulary, and natural
     contractions.
```

## F. Model and API Implementation Details

Voice clones are synthesized using ElevenLabs Professional Voice Cloning (PVC) with the eleven_monolingual_v1 model. For each of the 225 speakers, recordings from the Mozilla Common Voice Dataset are supplied to the PVC API, which adapts ElevenLabs' base TTS model to the target speaker identity.

Transcription of synthetic utterances uses three distinct ASR engines: Deepgram nova-2 (English), OpenAI gpt-4o-transcribe, and ElevenLabs scribe_v1. Each engine produces a separately stored transcript, and the three outputs are used independently throughout evaluation to expose provider-level variability.

## G. Text vs. Audio Input Modality Effects

This appendix section isolates the impact of input representation on function-calling accuracy by comparing native text inputs against audio-derived representations produced by ASR, while holding task content and function schemas fixed.

Across models and task types, we observe a consistent accuracy gap when transitioning from native text inputs to audio-based inputs. This degradation reflects a genuine modality shift rather than transcription error alone: audio inputs introduce disfluencies, prosodic variation, paraphrasing, and environmental noise that are absent in clean text. Even when user intent is preserved, these factors alter surface realizations in ways that interact poorly with schema-constrained function calling. Because function invocation requires precise argument structure and exact value matching, small deviations introduced by natural spoken language (such as restarts, reformulations, or detail dropping) are more likely to propagate into execution failures than to be absorbed by downstream reasoning. These results indicate that function calling remains substantially more brittle under realistic spoken interaction than under idealized text-only settings.

## H. Dataset Structure: Example

Each example stores: (i) the original clean intent/query, (ii) the speech-like rewritten transcript, (iii) transcript-level perturbation types and application order, (iv) clean audio and noisy audio filepaths, (v) acoustic effect types and intensity levels, (vi) three ASR transcripts (one per provider), and (vii) the function schema with callable functions, parameter types, and required arguments.

```
{
   "id": "live_simple_0-0-0",
   "question": [
      [
         {
            "role": "user",
            "content": "Can you retrieve the details for the user with the ID 7890,
               who has black as their special request?",
            "transcript": "Uh, get the, the details for user ID seven eight nine
               zero--the one with black as the special request.",
            "speech_like_features": [
               "sentence_restructuring",
               "disfluencies",
               "repetitions",
               "numbers_noise",
               "word_reordering",
               "simplified_verbs"
            ],
            "audio_path": "bfcl_eval/data/audio/BFCL_v4_live_simple/live_simple_0
               -0-0.mp3",
```

```
                "audio_path_noisy": "bfcl_eval/data/audio_noisy/BFCL_v4_live_simple/
                    live_simple_0-0-0.mp3",
                "noise_effects_applied": [
                    {
                        "name": "background_noise",
                        "intensity": "light"
                    },
                    {
                        "name": "volume_fluctuation",
                        "intensity": "light"
                    }
                ],
                "noise_processing_success": true,
                "asr_deepgram_audio_clean": "Get the the details for user ID seven eight
                    nine zero. The one with black is the special request.",
                "asr_openai_audio_clean": "Get the details for user ID 7890. The one
                    with black is the special request.",
                "asr_elevenlabs_audio_clean": "Uh, get the--the details for user ID 7890,
                    the one with black as the special request.",
                "asr_deepgram_audio_noisy": "Get the the details for user ID seven eight
                    nine zero. The one with black is the special request.",
                "asr_openai_audio_noisy": "Get the details for user ID 7890. The one
                    with black is the special request.",
                "asr_elevenlabs_audio_noisy": "Uh, get the- the details for user ID 7890.
                    The one with black as the special request."
            }
        ]
    ],
    "function": [
        {
            "name": "get_user_info",
            "description": "Retrieve details for a specific user by their unique
                identifier.",
            "parameters": {
                "type": "dict",
                "required": [
                    "user_id"
                ],
                "properties": {
                    "user_id": {
                        "type": "integer",
                        "description": "The unique identifier of the user. It is used to
                            fetch the specific user details from the database."
                    },
                    "special": {
                        "type": "string",
                        "description": "Any special information or parameters that need to
                            be considered while fetching user details.",
                        "default": "none"
                    }
                }
            }
        }
    ]
}
```

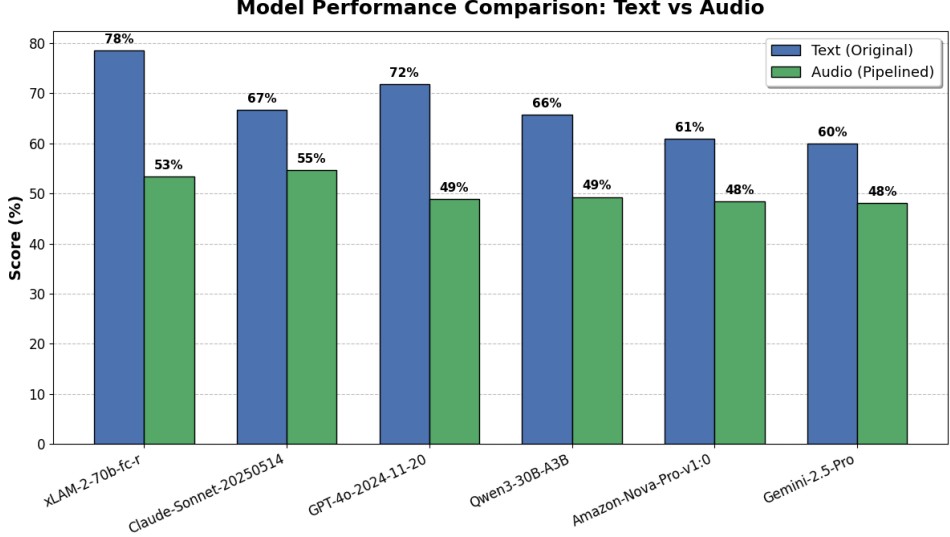

*Figure 8.* Overall comparison of function-calling accuracy under native text inputs versus ASR-derived transcripts (BFCL Text Audio). Replacing text with audio-derived representations consistently degrades performance across models, highlighting the brittleness of schema-constrained tool use to even minor transcription-induced shifts.

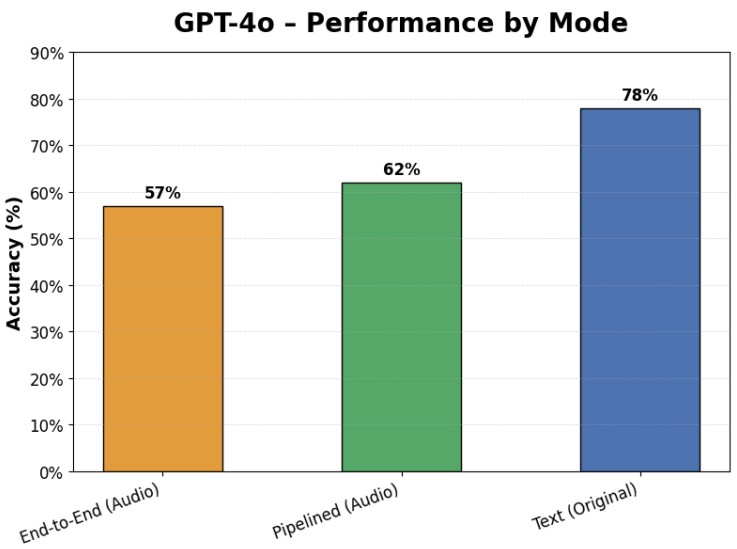

*Figure 9.* Text versus ASR-derived input performance for GPT-family models. Despite strong text-based tool-use capabilities, GPT models exhibit a systematic drop under ASR transcripts, reflecting sensitivity to entity normalization and formatting variations introduced upstream.

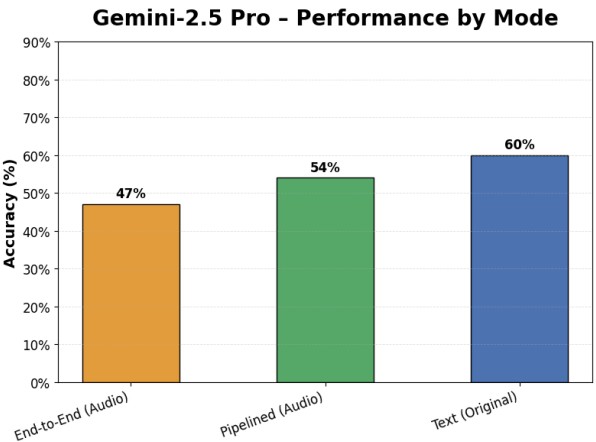

*Figure 10.* Text versus ASR-derived input performance for Gemini-family models. The observed degradation mirrors GPT trends, suggesting that ASR-induced representation shift is a model-agnostic bottleneck rather than an artifact of a specific architecture.

# I. Embedding-Space Analysis of Perturbation Effects

To complement our function-calling accuracy results, we analyze how speech and acoustic perturbations distort model representations. For text, we embed clean and perturbed queries using OpenAI's text-embedding-3-large (3072D). For audio, we extract intermediate encoder embeddings from Whisper-medium (1024D) via average pooling of final-layer representations. We compare clean versus perturbed inputs using Euclidean distance, cosine similarity, and canonical correlation analysis (CCA), reporting global averages, per-perturbation effects, and regression-based attribution controlling for co-occurring perturbations. A central question motivating this analysis is whether failures arise from global degradation of representations or from localized, direction-specific distortions.

**Euclidean Distance**    For text, embeddings from text-embedding-3-large are near unit norm, making Euclidean distance redundant with cosine similarity; we therefore report cosine similarity as the primary measure. For audio, embedding displacements are substantial and heterogeneous (mean 2.696, std 1.368). The largest shifts arise from reverb echo, mic rubbing, and reverb cave, followed by network and channel artifacts; background noise is omnipresent while mumbling has comparatively smaller effect. Ridge regression explains approximately 31% of variance. Crucially, these displacements occur without collapse of global structure (see CCA below), indicating anisotropic rather than uniform distortion and suggesting that downstream errors stem from direction-specific corruption of semantic components rather than wholesale representational failure.

**Cosine Similarity**    For text, global cosine similarity between clean and perturbed embeddings is 0.818, indicating meaningful misalignment. The largest drops arise from false starts, approximate quantifiers, vague references, and filepath verbalization, with filepath verbalization carrying the strongest partial effect under regression. These perturbations specifically disrupt named entities, quantities, and file paths, precisely the components that map to discrete function arguments, indicating that speech-like text does not uniformly degrade meaning but instead targets semantically precise components where even moderate angular drift leads to disproportionate errors in argument decoding.

For audio, embeddings remain tightly aligned globally (mean cosine similarity 0.995, std 0.006), but stratification reveals the largest angular drift for reverb echo and mic rubbing, followed by reverb cave and network effects. Regression confirms reverb echo ($-0.0069$) and mic rubbing ($-0.0038$) as dominant contributors. Together with the Euclidean results, this indicates that acoustic perturbations introduce subtle but structured directional drift, sufficient to impact downstream transcription and function-calling accuracy despite high overall similarity.

**Canonical Correlation Analysis**    CCA measures whether global embedding structure is preserved under perturbation by finding linear projections of clean and perturbed embeddings that maximize correlation, yielding a canonical correlation $\rho \in [0, 1]$. For text, pooled CCA shows near-perfect alignment ($\rho_1 \approx 0.9999$), confirming that the global latent subspace is preserved under speech-like perturbations. Per-feature stratification reveals lower alignment only for filepath verbalization ($\rho = 0.899$) and casual pronouns ($\rho = 0.901$), while all other features remain above 0.9997. For audio, pooled CCA is similarly tight ($\rho_1 \approx 1.0$), with worst-case effects from reverb echo and mic rubbing ($\rho = 0.9997$) and most conditions at or above 0.9998. This confirms that acoustic perturbations preserve principal embedding directions, and that the degradation observed in Euclidean and cosine metrics occurs in lower-variance, task-relevant subspaces.

**Summary**    Across all three metrics, a consistent picture emerges: global representation structure is preserved under perturbation (CCA), while localized shifts in lower-variance directions (cosine, Euclidean) disproportionately affect semantically precise components such as file paths, named entities, and quantities; exactly those required for correct discrete argument prediction. This explains why perturbations such as filepath verbalization and competing speech produce large downstream tool-calling failures despite minimal global misalignment, and provides an empirical basis for understanding how continuous embedding drift translates into discrete argument decoding errors.

