# OpenReview forum: "BFCL Audio: An Audio Function Calling Evaluation for Large Language Models"
_ICML.cc/2026/Conference — ICML 2026 regular_

### Official Review · Reviewer_op53 · 2026-03-09

**Soundness:** 3
**Presentation:** 2
**Significance:** 3
**Originality:** 3
**Overall Recommendation:** 4
**Confidence:** 4

**Summary:**

This paper introduces MFCL-Audio, a benchmark designed to evaluate audio-based tool calling for multimodal language models. The work highlights the gap between text-based tool use and speech-based interaction and proposes a dataset with controlled speech perturbations and noise conditions.

**Compliance With Llm Reviewing Policy:**

Affirmed.

**Final Justification:**

I understand the practical challenges of collecting large-scale real speech data, and the reported consistency between synthetic and real audio trends provides some support for the benchmark design. That said, I still believe the reliance on synthetic speech and the limited evidence on real-world validity somewhat constrain the overall impact. Therefore, I will maintain my original score.

**Key Questions For Authors:**

1. **Failure mode categorization**
   The paper introduces six failure modes (FM1–FM6), but it is unclear how these categories were defined and assigned. Could the authors clarify how errors were labeled and whether any annotation protocol or agreement was used?

2. **Synthetic speech realism**
   The dataset relies heavily on synthesized speech. Have the authors compared the synthesized audio with real human-recorded speech to validate whether the benchmark reflects realistic speech interaction conditions?

3. **Reproducibility and missing repository link**. The paper states that *“all preprocessing code, function schemas, and evaluation scripts are available publicly online in an anonymous repository.”* However, I could not find the repository link in the paper or the supplementary materials. Since the benchmark relies on specific data generation and evaluation procedures, providing access to this repository would be important for verifying reproducibility.

4. **Typos**

- **Terminology clarification (AST)**
   The paper uses the abbreviation **AST** in the evaluation section, but the full term  is not explicitly introduced when the abbreviation first appears.

- **Organization of Section 5**
   Section 5 contains only a single subsubsection (5.1). In this case, introducing a subsubsection level may be unnecessary and slightly disrupt the document structure. It might be clearer to merge this content directly into Section 5.

- **Unclear column definitions in Table 2**
   Table 2 includes columns such as **“Simple”, “Multiple”, and “Parallel”**, but their exact meanings are not clearly explained in the text. It would be helpful to briefly clarify what types of tool-calling structures these columns correspond to and how tasks are categorized.
- **Table 3 exceeds the page width**, which affects readability.
- **Figure 4 appears to contain an incomplete black artifact at the top**, suggesting that the figure may not have been cropped properly.

**Limitations:**

yes

**Strengths And Weaknesses:**

Strengths：

- Timely and relevant problem. The paper focuses on audio-based tool calling, an emerging capability for multimodal agents and voice-enabled assistants. Evaluating how speech inputs affect tool execution is an important and relatively underexplored problem.

- Comprehensive evaluation. The experiments evaluate multiple models under different conditions (text, clean audio, noisy audio) and provide analyses across several task categories. The inclusion of error analysis through failure modes is also useful for understanding model behavior.

- Potential usefulness for the community. If released with code and evaluation scripts, the benchmark could provide a useful resource for studying the robustness of audio-enabled agents and comparing different modeling approaches.

Weakness:

1. Unclear methodology for failure mode categorization. The paper introduces six failure modes (FM1–FM6), but the procedure used to define and assign these categories is not clearly described. Without a clear explanation of how errors are labeled or categorized, it is difficult to interpret the reported error distributions.

2. Reliance on synthetic speech data. The dataset is primarily constructed using synthesized speech. Although the authors conduct human evaluations of naturalness, the absence of comparisons with real human-recorded speech makes it difficult to assess how well the dataset reflects real-world speech interaction conditions.

3. Limited clarity in experimental presentation. Some experimental results are difficult to interpret due to insufficient explanation of certain table columns and evaluation categories. For example, the meanings of the “Simple”, “Multiple”, and “Parallel” columns in Table 2 are not clearly described.

4. Reproducibility concerns. The paper states that code and evaluation scripts are available in an anonymous repository, but no repository link is provided in the paper or supplementary materials.

---

> ### Author Rebuttal · Authors · 2026-03-31
>
> Thank you for the positive assessment and constructive suggestions. Because ICML rebuttal does not allow an updated PDF, the clarifications below are already reflected in our revised manuscript but are not visible in the current OpenReview version.
>
> ---
> ## Q1. Failure-mode categorization
> Failure modes are assigned only to failed benchmark items (a failed single-turn example or a failed multi-turn query). We built the six-way codebook from pilot inspection of failures, guided by the perturbation types in the audio-generation pipeline and the clarification protocol. Each failed interaction receives one primary label: the earliest error that causally explains the final incorrect outcome. FM4 is used when the assistant asks a disallowed or irrelevant clarification under the whitelist-based validator; FM5 when it produces conversational text instead of a valid tool call or allowed clarification. The remaining failed tool calls are labeled FM1/FM2/FM3/FM6 via a fixed rubric distinguishing intent contamination, premature execution, non-entity parameter corruption, and named-entity corruption. Six annotators independently labeled 1,000 overlapping failures and achieved 96% agreement; disagreements were adjudicated and the finalized rubric was then applied to the remaining failures.
>
> ---
> ## Q2. Synthetic speech realism
> We agree that perceptual naturalness alone does not validate real-speech robustness, so we additionally ran an internal validation using volunteer human recordings spanning different speakers, accents, tones, and background conditions. Across model families, the relative ranking and clean-vs-noisy trends were consistent with the synthetic benchmark, supporting that the pipeline captures realistic failure patterns. Due to privacy/approval constraints, we cannot release these recordings immediately, but we plan to include them in a future release.
>
> Furthermore, we have added a listening study that directly compares the synthetic audio with human ground truth speech. Please refer to our rebuttal response Q4 for reviewer uLP5 for more details.
>
> ---
> ## Q3. Reproducibility / repository link
> Code repo link:
> https://anonymous.4open.science/r/MFCL
>
>
> ---
> ## Q4. Presentation / typos
> “AST” refers to the Abstract Syntax Tree matching procedure used for single-turn evaluation. In the paper, single-turn scoring parses the predicted tool call into an AST and checks function-name and argument-level correctness without executing the function. We have expanded the term on first use in the revision.
>
> We have also merged Section 5.1 into Section 5, reformatted Table 3 to fit the page and recropped Figure 4 to remove the black artifact, and fixed the other formatting issues.
>
> ---
> ## Q5. Table 2 clarity
>
> Table 2 reports progressively harder tool-calling structures.
> The single-turn block reports *Simple* = one valid tool call; *Multiple* = choose among several tools; *Parallel* = repeated calls to the same tool; *Parallel Multiple* = multiple tools with repeated calls. It also separates *Expert Curated* examples, which systematically cover functional patterns, from *Community Sourced* examples, which are drawn from real-user-style inputs and are therefore noisier. The multi-turn block reports *Base* = standard sequences, *Miss Func* = no valid function exists, and *Miss Param* = missing arguments that should trigger clarification. *Relevance/Irrelevance* measure whether the model calls a tool only when warranted.
>
> These categories are important because audio-specific errors affect different parts of the pipeline differently: tool selection, argument grounding, call composition, and the decision to abstain or clarify. We have added an explicit paragraph before Table 2 clarifying these columns and task groupings.

---

> > ### Author Rebuttal · Reviewer_op53 · 2026-04-03
> >
> > I appreciate the authors’ efforts in addressing my concerns, particularly the clarification on failure-mode annotation and the additional validation with human recordings. I understand the practical challenges of collecting large-scale real speech data, and the reported consistency between synthetic and real audio trends provides some support for the benchmark design. That said, I still believe the reliance on synthetic speech and the limited evidence on real-world validity somewhat constrain the overall impact. Therefore, I will maintain my original score.

---

### Official Review · Reviewer_uLP5 · 2026-03-12

**Soundness:** 2
**Presentation:** 2
**Significance:** 4
**Originality:** 3
**Overall Recommendation:** 4
**Confidence:** 3

**Summary:**

This paper introduces a benchmark for evaluating LLMs called by audio functions. The authors curated a new dataset of 6.2K instructions with audio synthesized to resemble realistic conditions (spoken language with fillers, repetitions, spellings, etc.) of various speakers (identity, accent) under various acoustic environment conditions (background noise, cocktail party effect, packet loss). The audio is accompanied by transcripts from 3 different ASR systems. Then, the paper offers a deep analysis across many LLMs to identify instruction failures in relation to the introduced augmentation.

**Compliance With Llm Reviewing Policy:**

Affirmed.

**Final Justification:**

Initially, I had many concerns about this paper, which included reproducibility (no links/code provided, missing implementation details for the voice clones and ASR models, etc.); not-so-good presentation impacted by small, difficult to read figures and tables with missing explanations about the metrics used.

Furthermore, a crucial question I asked was focused on the naturalness evaluation of the simulated audio data, as these were not compared to any ground truth voice data to give a sense of the "realness" in it when compared to real-world recordings, as in real world scenarios all the LLMs must and will be listening to real world voices.

The authors have put a lot of effort into their rebuttal and cleared concerns surrounding the presentation and reproducibility. They also performed an additional listening study to strengthen the soundness of the presented dataset. The authors further acknowledged the limitations caused by their dataset being a simulated dataset as opposed to real-world recordings. This, however, still remains a limitation to the paper.

Nevertheless, I have decided to raise my final recommendation from "reject" to "weak accept", as the limitation still persists (although gracefully acknowledged and explained), and I cannot confirm the significance of all the presentation improvements (figures and tables) at this stage, which both prevent me from a clean "accept".

**Key Questions For Authors:**

1) In the dataset created, are you also storing information about the augmentations applied to each sample? How detailed is the dataset provided? It would be useful to provide a peek into it either in the paper or its appendix.

2) Can you confirm that each sample in the dataset is accompanied by three transcripts, each from a distinct ASR model? This is not fully clear from the text in Section 3.2.

3) Which ASR engines have you used? There is no reference, link, or name included in the paper. Can you please elaborate on this in the paper for increased reproducibility and soundness?

4) Which voice clone models have you trained? I was not able to find any information in the paper – no reference, name, link, or architecture described. Please update this information in the paper for increased reproducibility and soundness.

5) Regarding the listening test to assess naturalness of the synthesized audio, why has no ground truth reference been provided for the clean audio? This would increase the soundness of the paper by showing evidence of audio being close enough in naturalness to the GT samples, thus sounding "realistic".

Elaborating on these questions can improve the clarity and soundness of the paper and potentially improve my evaluation of the paper.

**Limitations:**

yes

**Strengths And Weaknesses:**

**Strengths:**
1) The dataset is constructed with a fine level of detail by applying many different augmentations to the prompts and audios alike.
2) The analysis of effects of different augmentations on the model's instruction output is thorough and offers many interesting insights.

**Weaknesses:**
1) Writing and presentation could use improvement. Figures and tables often contain text of very small font, and are difficult to read.
2) Some details of implementation and evaluation seem to be missing, e.g., information about the voice clone and ASR models deployed, description of Table 2 metrics, or all the parameter and function details for the creation of the dataset. This highly affects reproducibility in a negative way.
3) The listening test to assess the naturalness of the synthesized speech lacks ground truth reference. While the MOS scores look "fine", they do not give a full picture of how close the (clean) speech sounds to real recorded speech, thus putting a question mark over the "realistic conditions" aimed to achieve in the audio generation pipeline.
4) The Reproducibility Statement at the end of paper mentions code, scripts, and function schemas available, however, there is no link in the paper to support this.


**Detailed review:**

1) It is not clear what "voice clone" model was trained and used, as there is no link, reference, name, or architecture mentioned.

2) Same problem for the ASR models – there is no name, reference, link, or mention of how they were used/trained.

2) Inconsistencies in typography: Figures are sometimes referred to as "Fig.", and sometimes as "Figure". Please unify the notation.
Bold text at the start of paragraphs often represents a subtitle and is not a part of the sentence. But sometimes, e.g., in Section 4, in the first paragraph, the bold name starts the paragraph. Then, in the second paragraph, the bold name serves only as a subtitle and the full sentence starts after it, however, the subtitle is not separated by any dot, colon, or dash. I would kindly recommend sticking to a certain style and not properly separating sentences for increased clarity.

3) Presentation: Some figures and tables could use improvements for better readability. For example, figures 1, 2, 3, and 4 contain text of font smaller than the size of the font used for writing the paper, which is generally not recommended. (Figure 4 suffers the most) Table 2 is also unfortunately impacted by small font, but I understand it is difficult to make this better given its huge size, unless it is split in two. I suggest a revision of the affected figures and tables to improve readability by, e.g., utilizing the space a bit more efficiently.

4) Evaluation, results: Evaluation metrics in Table 2 are never explained anywhere. It is difficult to interpret the table without the knowledge of what each column means. This makes the table look irrelevant, unnecessary. I suggest the authors consider explaining the metrics (and results) in text, or move this table to the appendix and to give space to more (and discussed) results, e.g., you could use the gained space to improve (or enlarge some of) figures 1, 2, 3, and 4.

5) Table 3 - does *n* refer to WER? I suggest writing it out full for clarity. Also, percentage (relative) WER might be better than the number of occurrences for clarity (even if WER is not an important metric in that table).

---

> ### Author Rebuttal · Authors · 2026-03-31
>
> Thank you for the careful review. We agree that the submitted version omitted important implementation details and had presentation issues. Because ICML/OpenReview does not allow a revised PDF during rebuttal, the edits already made in Overleaf are not visible to reviewers, so we summarize the exact changes here.
>
> ---
> ## Q1. Reproducibility / implementation details
> The repo is at: https://anonymous.4open.science/r/MFCL
> . We also revised the text to make the pipeline explicit. Our “voice clones” are not an in-house TTS architecture: we use ElevenLabs Professional Voice Cloning (PVC). For each of 225 speakers, we provide Mozilla Common Voice speech to the PVC API, which adapts ElevenLabs’ base TTS model; synthesis uses eleven_monolingual_v1. For MFCL Text Audio, every audio sample is transcribed by three distinct ASR systems: Deepgram nova-2 (English), OpenAI gpt-4o-transcribe, and ElevenLabs scribe_v1, and each transcript is stored separately. We also reworded Section 3.1.3 to avoid implying that we trained a new voice-cloning architecture ourselves. Importantly, benchmark users do not need these services to run MFCL: we release the benchmark audio, all three transcripts, schemas, and evaluation code.
>
> ---
>
> ## Q2. Dataset structure
> Yes. Each example stores: (i) the original clean intent/query, (ii) the speech-like rewritten transcript, (iii) transcript-level perturbation types and application order, (iv) clean audio and noisy audio, (v) acoustic effect types and intensity levels, (vi) three ASR transcripts (one per provider), and (vii) the function schema with callable functions, parameter types, and required arguments. We added an example dataset record in the appendix.
>
> ---
> ## Q3. Presentation / tables
> We agree the presentation needed improvement. In the revision we enlarged Figs. 1–4, standardized figure references to “Fig.” throughout, fixed inconsistent bolded subtitles, split Table 2 into expert-curated and community-sourced sections for readability, and relabeled Table 3’s “n” as “Samples Evaluated.” We added WER (%) to Table 3 to make clear that WER is only a supplementary sanity check and does not track tool-use reliability well. “Samples Evaluated” denotes the number of utterances per condition. Larger counts for background noise reflect multiple noise types and intensities, while isolated conditions like mic rubbing or reverb cave have smaller sets. Intent preservation and confidence are computed per condition, so comparisons remain valid despite unequal group sizes. For instance, mic rubbing shows near-zero intent loss despite non-trivial WER, whereas competing speech causes severe intent loss at similar or lower WER.
>
> For the expanded evaluation metrics for Table 2, please see our rebuttal comment Q3 for reviewer op53.
>
> ---
> ## Q4. Naturalness evaluation
> We agree that MOS alone does not measure closeness to real speech. To address this, we ran an additional matched pairwise CMOS study on the clean condition against real human recordings of the same rewritten transcripts used by our generation pipeline: 20 raters, 100 matched pairs, 2,000 total judgments. Ground-truth speech was preferred in 58% of trials, MFCL clean audio in 12%, and 30% were ties/no preference, with mean CMOS +0.68 in favor of ground truth. These results have already been added in the revised manuscript. Our goal is not to make MFCL indistinguishable from human recordings, but to provide a perceptually realistic clean-speech condition for benchmarking audio function calling; this paired evaluation more directly supports that claim.

---

> > ### Author Rebuttal · Reviewer_uLP5 · 2026-04-04
> >
> > Dear Authors,
> >
> > Thank you for your elaborate response. Find my responses to the points below:
> >
> > **Q1:** Thank you for sharing the repository and for specifying the voice clones and ASR implementations. Please make sure this additional information is present in the final manuscript (be it in the main text, or an implementation section of your appendix).
> >
> > **Q2:** Thank you for elaborating on the dataset structure and expanding the manuscript.
> >
> > **Q3:** Good job improving all the presentation related details and for clarifying the metrics in Table 2.
> >
> > **Q4:** Thank you for expanding this with another listening study. This is useful and improves the soundness. Nevertheless, I still think we should keep in mind the realness of audio as a whole to be the desired attribute of the data for your study, as that would be representative of real world scenarios. Therefore, I kindly suggest acknowledging, as part of your limitations, that real world data (no voice clones, but recordings, manual transcripts, and real situations such as office environment, background noises, etc.) would still be an improvement and that the simulations performed to obtain your current data can’t fully reproduce the real-world character.
> >
> > Most of my concerns have been addressed well. I would welcome a short statement to my point about Q4, but for now, I do not have further questions and deem the rebuttal as resolved. I will re-evaluate your paper shortly.
> >
> > Good luck!

---

> > > ### Author Response · Authors · 2026-04-05
> > >
> > > Dear Reviewer uLP5,
> > > Thank you for this suggestion. We agree and will state this explicitly as a limitation in the revised paper. While MFCL-Audio is designed to approximate deployment-relevant speech and acoustic conditions in a controlled and reproducible way, it remains a synthetic benchmark. In particular, voice-cloned speech, rewritten transcripts, and simulated acoustic perturbations cannot fully capture the spontaneity, recording variability, and environmental complexity of naturally recorded real-world interactions. We will therefore add a limitation noting that benchmarks built from real recordings, manual transcripts, and naturally occurring settings (e.g., office environments and authentic background noise) would provide stronger ecological validity and would be an important direction for future work. We view MFCL-Audio as a controlled diagnostic benchmark that complements, rather than replaces, evaluation on fully real-world data.

---

### Official Review · Reviewer_L9pB · 2026-03-13

**Soundness:** 3
**Presentation:** 2
**Significance:** 2
**Originality:** 3
**Overall Recommendation:** 4
**Confidence:** 4

**Summary:**

The paper introduces MFCL-Audio, a large-scale benchmark designed to evaluate end-to-end audio function calling (tool use) in Large Language Models. The benchmark comprises 6.2K expert-verified tasks divided into two suites: MFCL Text Audio (evaluating pipelined ASR-to-LLM systems) and MFCL True Audio (evaluating direct audio-in end-to-end models). To create realistic test conditions, the authors developed a controllable audio generation pipeline that injects spontaneous speech disfluencies (e.g., self-corrections, hesitations) via an LLM controller, renders the text using diverse voice clones, and applies real-world acoustic augmentations like background noise and network cuts. Evaluation is performed using an automated, judge-free grading system relying on AST-based matching for single-turn calls and state/response verification for multi-turn interactions. The authors propose a six-part failure mode taxonomy (e.g., Intent Blending, Clarification Misfires) and conduct ablations showing that competing speech and discourse-altering disfluencies are the primary drivers of function-calling failures.

I feel the paper is around the borderline when the functions and empirical sides are interesting, less theoretical or previous linguistic perspectives has been studied or connected. I will slightly revise my score up or down depends on the authors' feedbacks.

**Compliance With Llm Reviewing Policy:**

Affirmed.

**Key Questions For Authors:**

- Would you consider to formalize the failure modes (particularly regarding filepath verbalization and symbolic reasoning) using representation learning theory? Drawing connections to frameworks like SpeechIQ (ACL 2025) to explain the misalignment between continuous acoustic spaces and discrete schema constraints etc.

- For end-to-end models, have you investigated how much the latent embeddings drift when subjected to the "competing speech" augmentation compared to "reverb"? A brief analysis of embedding distances could explain the stark difference in entity confusion versus word deletion.

- How does the AST-based grader handle cases where a model predicts a functionally equivalent but syntactically different argument structure? The paper mentions type-aware normalization, but further detail on edge cases would aid reproducibility.

**Limitations:**

yes.

**Strengths And Weaknesses:**

Pros

- The controllable pipeline for generating speech-like transcripts is good. By using an LLM to inject mutually exclusive classes of disfluencies while maintaining the validity of the underlying function schema.

- The proposed failure-mode taxonomy provides excellent granularity. Identifying that models fail not just from WER but from phenomena like "Conversational Drift" (where models revert to standard dialog instead of outputting JSON) or "Clarification Misfires.

Cons

- While the empirical ablations are thorough, the paper fundamentally lacks a theoretical perspective on why these models fail from a representation learning standpoint. For ICML, an empirical benchmark is vastly strengthened when it diagnoses the geometric or information-theoretic breakdowns in the latent space. For example, recent major work like SpeechIQ (ACL 2025) provides frameworks for evaluating the intrinsic intelligence and alignment of speech representations. The authors note that end-to-end models struggle significantly with "filepath verbalization" due to a mismatch between natural speech and discrete string arguments. This could be formalized by analyzing the mutual information $I(\mathbf{z}_{acoustic}; \mathbf{z}_{semantic})$ between the acoustic embeddings and the target symbolic token space. Without this theoretical lens, the paper remains somewhat descriptive rather than explanatory.

- The paper relies heavily on downstream task success (AST matching) and basic intent preservation. It would benefit from intermediate representation metrics to quantify how much specific acoustic augmentations (like reverb vs. competing speech) perturb the latent embeddings before they decode into function calls.

---

### Decision · Program_Chairs · 2026-04-30

**Decision:**

Accept (regular)

**Comment:**

This paper introduces a new benchmark for evaluating LLMs on audio-based instructions, including a carefully constructed dataset with diverse augmentations and a thorough analysis of model behavior under realistic speech variations. The paper's main contribution is valuable and timely, and the empirical analysis provides useful insights into failure modes across models.

Reviewers raised concerns regarding reproducibility, clarity of evaluation metrics, and presentation quality, as well as the realism of the synthesized audio due to limited grounding against real recordings. The authors addressed most of these concerns during the rebuttal phase, including additional evaluations and additional clarifications.

Although the reliance on synthetic data remains a limitation, overall, the work represents a meaningful contribution, hence I recommend acceptance.